# Testing Denmark: a Danish Nationwide Surveillance Study of COVID-19

Kamille Fogh,[a,b,q] Jarl E. Strange,[a,q] Bibi F. S. S. Scharff,[c,q] Alexandra R. R. Eriksen,[a,b,q] Rasmus B. Hasselbalch,[a,b,q] Henning Bundgaard,[d,q] Susanne D. Nielsen,[e,q] Charlotte S. Jørgensen,[f] Christian Erikstrup,[f,g] Jakob Norsk,[a,b,q] Pernille Brok Nielsen,[a,b,q] Jonas H. Kristensen,[a,b,q] Lars Østergaard,[f,g] Svend Ellermann-Eriksen,[f,h] Berit Andersen,[f,i] Henrik Nielsen,[j,s] Isik S. Johansen,[k,t] Lothar Wiese,[l] Lone Simonsen,[m] Thea K. Fischer,[n,v] Fredrik Folke,[a,o,q] Freddy Lippert,[o,q] Sisse R. Ostrowski,[c,q] Thomas Benfield,[p,q] Kåre Mølbak,[f,u] Steen Ethelberg,[f,v] Anders Koch,[e,f,q] Ute Wolff Sönksen,[f] Anne-Marie Vangsted,[f] Tyra Grove Krause,[f] Anders Fomsgaard,[f] Henrik Ullum,[f] Robert Skov,[f] Kasper Iversen[a,b,q]

aDepartment of Cardiology, Copenhagen University Hospital, Herlev and Gentofte, Denmark
bDepartment of Emergency Medicine, Copenhagen University Hospital, Herlev and Gentofte, Denmark
cDepartment of Clinical Immunology, Copenhagen University Hospital, Rigshospitalet, Denmark
dDepartment of Cardiology, Copenhagen University Hospital, Rigshospitalet, Denmark
eDepartment of Infectious Diseases, Copenhagen University Hospital, Rigshospitalet, Denmark
fStatens Serum Institut, Copenhagen, Denmark
gDepartment of Infectious Diseases, Aarhus University Hospital, Aarhus, Denmark
hDepartment of Clinical Microbiology, Aarhus University Hospital, Aarhus, Denmark
iUniversity Research Clinic for Cancer Screening, Randers Regional Hospital, Randers, Denmark
jDepartment of Infectious Diseases, Aalborg University Hospital, Aalborg, Denmark
kDepartment of Infectious Diseases, Odense University Hospital, Odense, Denmark
lDepartment of Infectious Diseases, Zealand University Hospital, Roskilde, Denmark
mDepartment of Science and Environment, Roskilde University, Roskilde, Denmark
nDepartment of Clinical Research, North Zealand Hospital, Hillerød, Denmark
oCopenhagen Emergency Medical Services, Copenhagen, Denmark
pDepartment of Infectious Diseases, Copenhagen University Hospital, Amager and Hvidovre, Hvidovre, Denmark
qDepartment of Clinical Medicine, University of Copenhagen, Copenhagen, Denmark
rDepartment of Clinical Medicine, Aarhus University, Aarhus, Denmark
sDepartment of Clinical Medicine, Aalborg University, Odense, Denmark
tDepartment of Clinical Research, University of Southern Denmark, Odense, Denmark
uDepartment of Veterinary and Animal Sciences, University of Copenhagen, Frederiksberg C, Denmark
vDepartment of Public health, University of Copenhagen, Copenhagen, Denmark

**ABSTRACT** "Testing Denmark" is a national, large-scale, epidemiological surveillance study of SARS-CoV-2 in the Danish population. Between September and October 2020, approximately 1.3 million people (age >15 years) were randomly invited to fill in an electronic questionnaire covering COVID-19 exposures and symptoms. The prevalence of SARS-CoV-2 antibodies was determined by point-of care rapid test (POCT) distributed to participants' home addresses. In total, 318,552 participants (24.5% invitees) completed the study and 2,519 (0.79%) were seropositive. Of the participants with a prior positive PCR test ($n = 1,828$), 29.1% were seropositive in the POCT. Although seropositivity increased with age, participants 61 years and over reported fewer symptoms and were tested less frequently. Seropositivity was associated with physical contact with SARS-CoV-2 infected individuals (risk ratio [RR] 7.43, 95% CI: 6.57–8.41), particular in household members (RR 17.70, 95% CI: 15.60–20.10). A greater risk of seropositivity was seen in home care workers (RR 2.09, 95% CI: 1.58–2.78) compared to office workers. A high degree of adherence with national preventive recommendations was reported (e.g., >80% use of face masks), but no difference were found between seropositive and seronegative participants. The

Address correspondence to Kamille Fogh, kamille.fogh.01@regionh.dk.

The authors declare no conflict of interest.

seroprevalence result was somewhat hampered by a lower-than-expected performance of the POCT. This is likely due to a low sensitivity of the POCT or problems reading the test results, and the main findings therefore relate to risk associations. More emphasis should be placed on age, occupation, and exposure in local communities.

**IMPORTANCE** To date, including 318,522 participants, this is the largest population-based study with broad national participation where tests and questionnaires have been sent to participants' homes. We found that more emphasis from national and local authorities toward the risk of infection should be placed on age of tested individuals, type of occupation, as well as exposure in local communities and households. To meet the challenge that broad nationwide information can be difficult to gather. This study design sets the stage for a novel way of conducting studies. Additionally, this study design can be used as a supplementary model in future general test strategy for ongoing monitoring of COVID-19 immunity in the population, both from past infection and from vaccination against SARS-CoV-2, however, with attention to the complexity of performing and reading the POCT at home.

**KEYWORDS** COVID-19, SARS-CoV-2, population study

National seroprevalence data on antibodies to SARS-CoV-2 can guide national health policies in understanding transmission routes of COVID-19 pandemic (1, 2) but, a large sample size is required to describe the spread of infection, risk factors, and severity of the infection (3).

Denmark has 5.8 million inhabitants and as of July 30, 2021, there have been more than 316,068 (5%) confirmed cases of infection and more than 2,548 COVID-19 related deaths in Denmark (4). The epidemic has been characterized by two infection waves: spring 2020 and autumn-winter 2020/2021. Two lockdowns were imposed by the government: March 11 to April 15, 2020, and December 17, 2020, to February 8, 2021 (5). Testing for SARS-CoV-2 by PCR was established in March 2020. From March 12, 2020, individuals with moderate to severe symptoms of respiratory tract infection were offered testing. From April 21, 2020, testing was available for individuals with mild symptoms and asymptomatic contacts, and since May 18, 2020, nationwide high-intensity, free of charge testing for SARS-CoV-2 infection has been performed using PCR (6). Vaccination against COVID-19 began on December 27, 2020, with residents and employees at nursing homes and frontline staff at hospitals (7).

The seroprevalence has been reported for selections of the Danish population in summer and autumn 2020 with estimates of seroprevalence of approximately 2.0% (5, 8–12), but hitherto no national investigation at this scale has been performed in Denmark.

The study "Testing Denmark" was a nationwide surveillance study of SARS-CoV-2 infection in the Danish population, launched in September 2020.

The aim of this study was to explore possible risk factors for seropositivity by questionnaire data and to estimate the seroprevalence of SARS-CoV-2 antibodies among Danish citizens.

## RESULTS

**Baseline variables and association with seropositivity.** In total, 474,411 participants (36.5% of invitees) replied to the electronic questionnaire and 397,843 received a POCT between October 2 and October 11, 2020. Invited persons who did not answer the questionnaire were more often males and with lower participation among persons aged <35 and >74 years of age with no noticeable geographical variations (Table S2). Participants not providing POCT results were more often younger with no noticeable geographical variations (Table S2). The final study population comprised 318,552 participants (24.5% invitees) who answered the questionnaire and provided the results of the POCT (Fig. 1). Age and sex distribution of the study population are shown in Fig. S1.

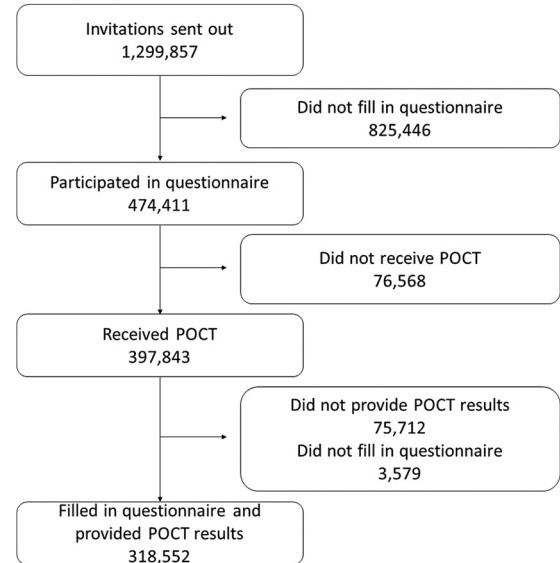

**FIG 1** CONSORT diagram.

A total of 2,519/318,552 (0.79%) participants tested seropositive with 852 (0.27%) participants being positive for IgG antibodies, 1,078 (0.34%) for IgM antibodies, and 589 (0.18%) positive for both IgG and IgM antibodies. The seroprevalence increased with age with a higher proportion of IgM positive compared to IgG positive (age group 15–30: 0.24% IgG positive, 0.26% IgM positive, age group >61–75: 0.25% IgG positive, 0.38% IgM positive, Fig. S2). No clear difference was found between IgG and IgM according to age groups (data not shown). For IgG, 9,294 (2.92%) and for IgM 9,269 (2.91%) were inconclusive, respectively.

The seroprevalence was statistically significantly lower for males. However, the clinical difference was minor (Table 1 and Fig. S3). The comorbidity burden was higher in seropositive participants and reached statistical significance for participants with hypertension, stroke, diabetes, and chronic obstructive pulmonary disease (COPD). There was a numerically higher proportion of seropositive females among participants smoking >10 cigarettes per day and among participants consuming >21 standard drinks of alcohol per week. For body-mass index (BMI), the proportion of seropositive females was higher in the category underweight or obese; see Fig. S4.

**POCT findings in participants with previous COVID-19, or a positive PCR.** When comparing self-estimated risk of infection with POCT results, only 0.5% of participants who self-estimated no prior infection were seropositive. Contrary, 13.5% of participants, who thought they had been infected, were seropositive. In comparison, 29.1% of participants, who had previously tested positive on a PCR test were seropositive (Fig. S5 and Fig. S6). For time between positive PCR test and POCT, 37.7% were seropositive 21–30 days after the PCR test. For seropositive participants with an available date of positive POCT and of the PCR test ($n = 804$), the proportion of seropositive participants decreased with increasing time between PCR test and POCT (Fig. S7). Differences between seropositive and seronegative who had previously tested positive on PCR test are shown in Table 2. Notably, time between positive PCR and POCT was lower for seropositive than seronegative participants.

Figure S8 show geographical variations between municipalities in seropositivity and variations in population density.

**Risk factors for seropositivity.** Most participants followed multiple recommended public health measures to prevent infection, e.g., >80% reported use of face masks. However, when examining serostatus according to behavior, no difference in serostatus was found between individual protective health measures, e.g., 82.5% of seronegative and 84% of seropositive reported use of face masks (Fig. 2).

Participants who had physical contact or lived in a household with a SARS-CoV-2

**TABLE 1** Baseline characteristics of the study cohort on sex, age, BMI, smoking, alcohol use, previous test result, and comorbidities stratified by seropositivity[a]

| Full cohort | Seronegative | Seropositive | P |
|---|---|---|---|
| n | 316,033 | 2,519 | |
| Age (yrs) (median [IQR]) | 53 [39-64]) | 55 [42-64] | 0.041 |
| Male (%) | 113,412 (422) | 1,012 (40.2) | <0.001 |
| Body mass index (median [IQR]) | 25.4 [22.8, 28.7] | 25.5 [23, 29] | 0.115 |
| Ever smoker (%) | 168,024 (53.2) | 1,375 (54.6) | 0.161 |
| Alcohol use (%) | 36,747 (12.9) | 302 (13.5) | 0.443 |
| | | | |
| Comorbidities (%) | | | |
| Myocardial infarction | 6562 (2.1) | 59 (2.3) | 0.389 |
| Stroke | 9067 (2.9) | 91 (3.6) | 0.030 |
| Hypertension | 82215 (26.0) | 711 (28.2) | 0.013 |
| Diabetes | 17528 (5.5) | 165 (6.6) | 0.032 |
| Cancer | 23250 (7.4) | 185 (7.3) | 1.000 |
| Rheumatoid arthritis | 19309 (6.1) | 176 (7.0) | 0.074 |
| COPD | 13872 (4.4) | 150 (6.0) | <0.001 |
| Asthma | 43996 (13.9) | 375 (14.9) | 0.172 |
| Other chronic disease | 56134 (17.8) | 456 (18.1) | 0.675 |
| | | | |
| Work type | | | |
| Not working | 123,959 (39.2) | 947 (37.6) | |
| Office work | 83,401 (43.4) | 538 (34.2) | |
| Tradesman | 20,653 (10.8) | 154 (9.8) | |
| School/other education | 23,773 (12.4) | 199 (12.7) | |
| Shop work | 9,103 (4.7) | 78 (5.0) | |
| Nursing home | 5,768 (3.0) | 57 (3.6) | |
| Healthcare sector | 21,863 (11.4) | 287 (18.3) | |
| Home care | 3,827 (2.0) | 52 (3.3.) | |
| Other | 44,755 (23.3) | 370 (23.5) | |
| | | | |
| Exposed to SARS-CoV-2 infected person | | | |
| Yes | 32,099 (10.2) | 713 (28.3) | |
| No | 212,966 (67.4) | 1,208 (48.0) | |
| Do not know | 70,968 (22.5) | 598 (23.7) | <0.001 |

[a]Alcohol use: Reporting >7 units of alcohol a week for females or >14 units of alcohol for male. The cohort enncompasses students, stay-at-home persons, out of job, long-term sick leave, retired, and persons on parental leave. Occupations are counted as the percentage of seropositive among those working. Each participant can have more than one type of occupation; thus, the percentage sums up to more than 100.

infected person had the highest risk of being seropositive compared to participants who reported not being exposed to a SARS-CoV-2 infected person; RR of 7.43 (95% CI: 6.57 to 8.41) and 17.70 (95% CI: 15.60 to 20.10), respectively (Fig. 3). Among participants exposed to an infected person within the household, the proportion of seropositive participants was higher in smaller households (see Fig. S9). However, when adjusting for sex, age, and household size, there was no significant increased risk for lower household size and risk of seropositivity (Table 3).

Among professionals (full-time, part-time, and self-employed), working in the health care sector or with home care was associated with a higher risk of seropositivity compared to office work; health care sector: RR 2.02 (95% CI: 1.75 to 2.33), home care: RR 2.09 (95% CI: 1.58 to 2.78), see Fig. 4.

**Symptoms.** For individual symptoms, loss of taste and smell were associated with the highest risk of being seropositive: ageusia (RR 5.91, 95% CI: 5.41 to 6.46) and anosmia (RR 4.84, 95% CI: 4.43 to 5.29). The risk of seropositivity for each symptom is shown in Fig. 5.

Participants in advanced age groups had experienced less symptoms compared to participants in younger age groups with 39.5% in age group >75 years compared to 8.2% in the age group 15–30 years experiencing no symptoms (Fig. S10). Participants in advanced age groups had been tested fewer times compared to participants in younger age groups irrespective of sex (Fig. S11).

**TABLE 2** Characteristics of the study cohort who previously testes positive on PCR test

| Full cohort | Seronegative | Seropositive | Total | P |
|---|---|---|---|---|
| n | 1,296 | 532 | 1,828 | |
| Age (yrs) (median [IQR]) | 47 [31-59]) | 51 [40−61] | 49 [34−59] | <0.001 |
| Male (%) | 480 (37.0) | 233 (43.8) | 713 (39.0) | 0.008 |
| Body mass index (median [IQR]) | 24.9 [22.4, 28.4] | 25.6 [23.0, 29.1] | 25.1 [22.6, 28.7] | 0.003 |
| Days between pos. PCR and POCT (median [IQR]) | 58 [26, 188] | 38 [23, 176] | 46.5 [25, 187] | 0.082 |
| Missing[a] | 693 | 331 | 1,024 | |
| | | | | |
| Comorbidities (%) | | | | |
| Myocardial infarction | 26 (2.0) | 11 (2.1) | 37 (2.0) | 1.000 |
| Stroke | 31 (2.4) | 17 (3.2) | 48 (2.6) | 0.415 |
| Hypertension | 257 (19.8) | 129 (24.2) | 386 (21.1) | 0.041 |
| Diabetes | 67 (5.2) | 38 (7.1) | 105 (5.7) | 0.124 |
| Cancer | 75 (5.8) | 33 (6.2) | 108 (5.9) | 0.815 |
| Rheumatoid arthritis | 72 (5.6) | 31 (5.8) | 103 (5.6) | 0.907 |
| COPD | 46 (3.5) | 21 (3.9) | 67 (3.7) | 0.784 |
| Asthma | 202 (15.6) | 84 (15.8) | 286 (15.6) | 0.970 |
| Other chronic disease | 211 (16.3) | 84 (15.8) | 295 (16.1) | 0.850 |
| Alcohol use (%) | 144 (12.5) | 56 (11.7) | 22 (12.3) | 0.708 |
| Ever smoker (%) | 607 (46.8) | 278 (52.3) | 885 (48.4) | 0.040 |

[a]Missing encompasses participants who did not have an available date of both positive PCR and POCT. Thus, days between positive PCR and POCT could not be calculated for these participants.

## DISCUSSION

To our knowledge this is the largest population-based SARS-CoV-2 surveillance study performed. The main findings can be summarized as follows; females had a higher seroprevalence than males. Elderly participants were more often seropositive despite fewer symptoms and less often PCR tests. The study showed a high degree of adherence with national recommendation but no clear difference in reported compliance between seropositive and seronegative participants in the study period. Unexpected a prevalence of SARS-CoV-2 antibodies of only 0.79% was reported and was lower than other seroprevalence studies performed in the same time interval (5, 13). Accordingly, only 29% of PCR positive were POCT seropositive in our study. The low seroprevalence at 0.79% in our study may be due to low sensitivity of the POCT used or due to difficulties in reading the test results, since 2.9% were inconclusive. POCT in general have a lower diagnostic performance compared to laboratory testing (14) and the Livzon POCT have been found to have a lower-than-expected sensitivity (15, 16). Test results also depend on the prevalence of infection in the population which will be low when screening asymptomatic and higher for those with suggestive symptoms. In low prevalence settings, true positive test results are uncommon. As such, the predictive value of a positive test will be lower in individuals with a low background risk of infection (17). Only 0.5% of the Danish population were confirmed PCR positive during the study period. The diagnostic testing window is also of importance as the study was performed 7 to 8 months after the first COVID-19 case in Denmark, as studies have shown waning antibody levels within several months after infection (18). From September 2020 the incidence of infected people in Denmark increased, peaking in December 2020. This could explain the higher proportion of IgM positive than IgG positive found in this study. The first infection wave in spring 2020 was minor, fewer were therefore infected back then, resulting in fewer with IgG antibodies and more with IgM antibodies during the study period (19). The antibody response of IgM and IgG is found to be highest about 2–4 weeks after symptom onset and decrease afterwards (14). 37% of our study participants had a positive POCT 20–30 days after a positive PCR. In addition, we found that for seronegative, longer time had passed from a previously positive PCR test than for seropositive. Importantly, inconclusive tests were treated as negative in our study, and weak lines suggesting a positive test result, could be misinterpreted as a negative test result. In other Danish studies, the tests (POCT and ELISA) have been performed and read or analyzed by professional staff which increases

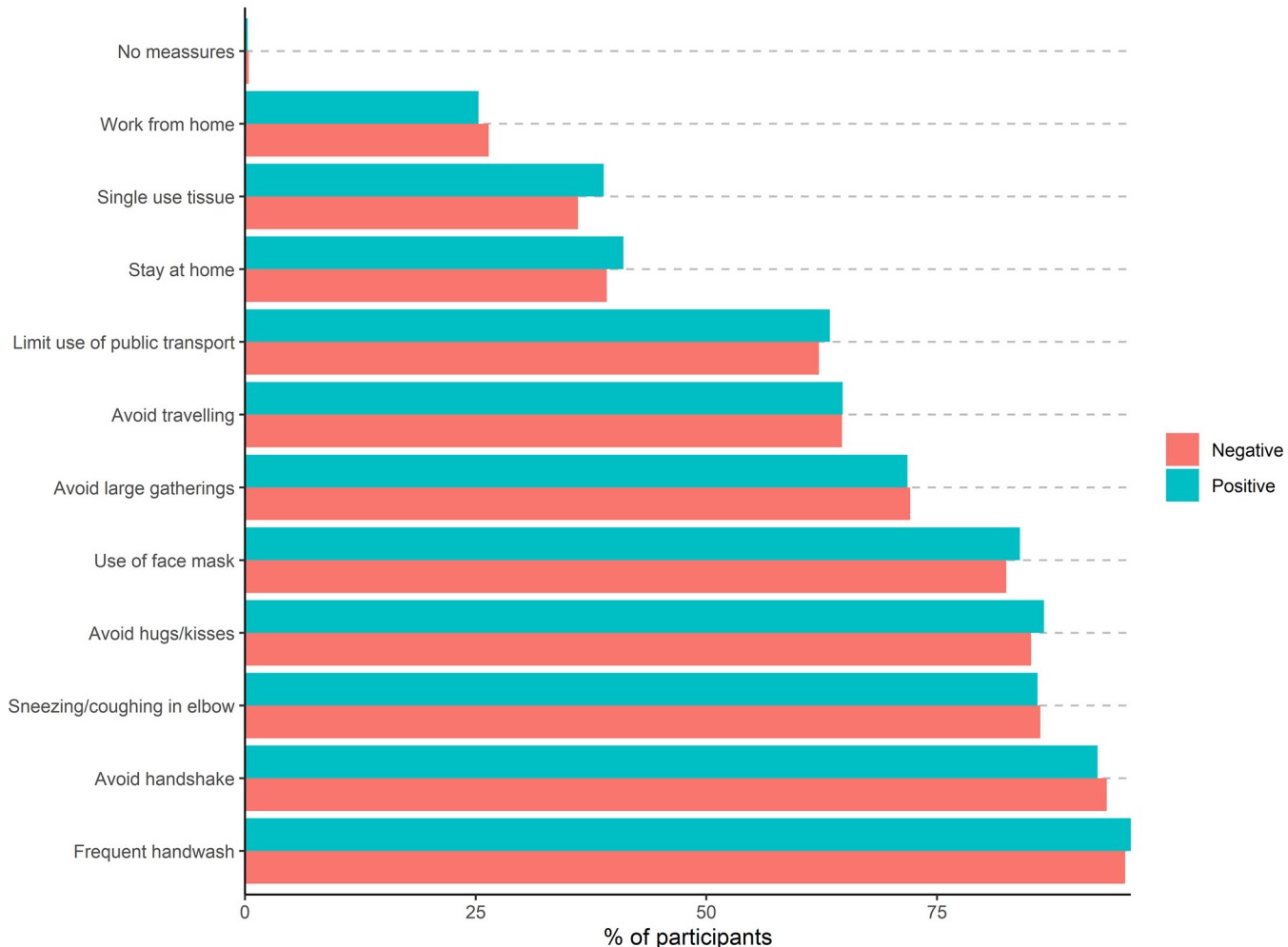

**FIG 2** Proportion of participants following public health measures stratified for serostatus among 318,552 individuals.

the performance of the test. Consequently, the seroprevalence is likely underestimated in our study. However, seropositivity was low among participants who did not have a previous positive PCR test, indicating a high specificity of the POCT, thus the associations found are reliable.

**Age and sex.** Until October 2020, 2·4 million people in Denmark had been tested with PCR, and 27,998 people were confirmed PCR positive (0.5% of the total population) (4). A population-based study in Denmark with 7,015 participants from August 2020 found a seroprevalence of 2.0% (age >12 years) measured by Wantai SARS-CoV-2 Ab ELISA (5), the point estimates tended to be higher in the age group 18–39 years and lower in the age group >65 years, with no difference observed by sex. A convenience sample of blood donors tested in October 2020 with ELISA found a seroprevalence of 2.1% (adults aged 18–70) (13). In contrast, we found a seroprevalence of only 0.79%, with the highest proportions of seropositivity among older participants and females. The discrepancy between the estimates in this study and those mentioned in earlier Danish studies may partly be due to test performance and the selection of participants. As mentioned, the earlier Danish studies were performed and analyzed by professional staff and the participants were from selected groups.

A Danish study of household transmission, with individual level register data on all national PCR test for SARS-CoV-2 for the period February-July 2020, suggested that susceptibility to infection increases with age (20). Other international studies tends to show trends in line with our results with increasing seropositivity with age (21) and females having increased IgG positivity (22). Sending our test material to participants

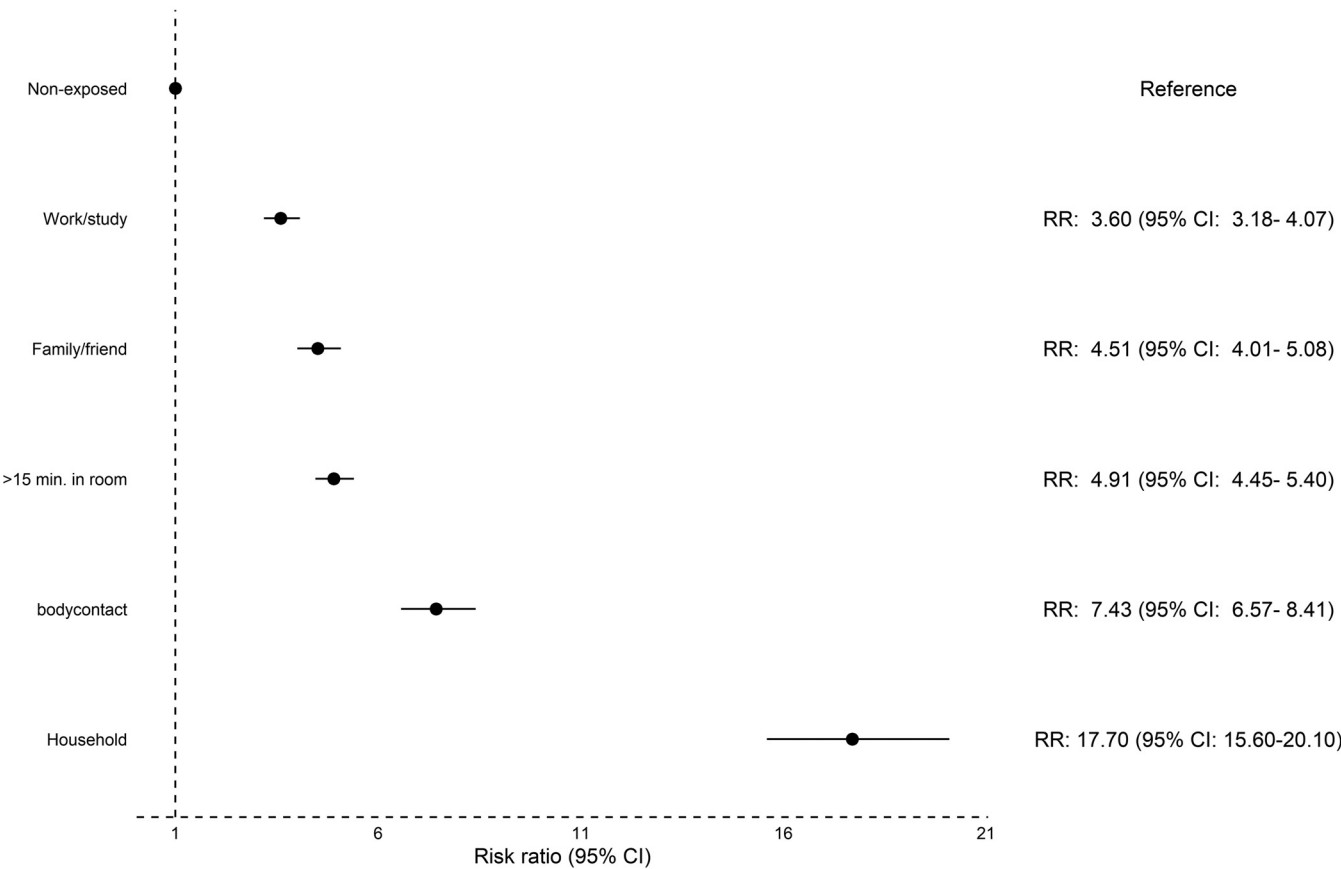

**FIG 3** Risk ratio for seropositivity in 32,812 participants exposed to COVID-19 infected persons in various settings. For each setting, participants exposed to COVID-19 infected persons was compared to participants not exposed in this setting (reference group).

at home for self-test may have prevented participation of vulnerable and/or older people susceptible to infection, as the test-setup required online access to read the invitation by e-Boks as well as sending answers to the questionnaire and POCT result. The complexity of performing and reading the POCT could also have been a factor in low participation rate among participants in the older age group. This is supported by our findings that participation in POCT was high in all age groups except the young and could partially explain the difference in seroprevalence between our study and aforementioned Danish studies, which included healthy blood donors as well as a population that should attend a venous blood sample.

**Testing and symptoms.** Elderly participants reported fewer previous tests. Compared to younger participants, elderly participants might have fewer social contacts and/or could have isolated themselves more thus avoiding potential close contact with infected persons. Younger participants may be more exposed to infection by having more social contacts or via their employment, and it should be noted that individuals in the working age who were unable to work from home may attend PCR testing more often than people

**TABLE 3** Odds ratio for age, sex, and household size stratified by seropositivity of the cohort

| Variable | | Odds ratio | 95% CI | P value |
|---|---|---|---|---|
| Age | | 1.02 | [1.01;1.03] | <0.001 |
| Male | | 1.01 | [0.77;1.34] | 0.920 |
| Household | 2 | Ref | | |
| | 3 | 0.75 | [0.51;1.09] | 0.128 |
| | 4 | 0.73 | [0.50;1.07] | 0.106 |
| | 5 | 0.58 | [0.34;1.01] | 0.054 |
| | >5 | 0.59 | [0.30;1.16] | 0.127 |

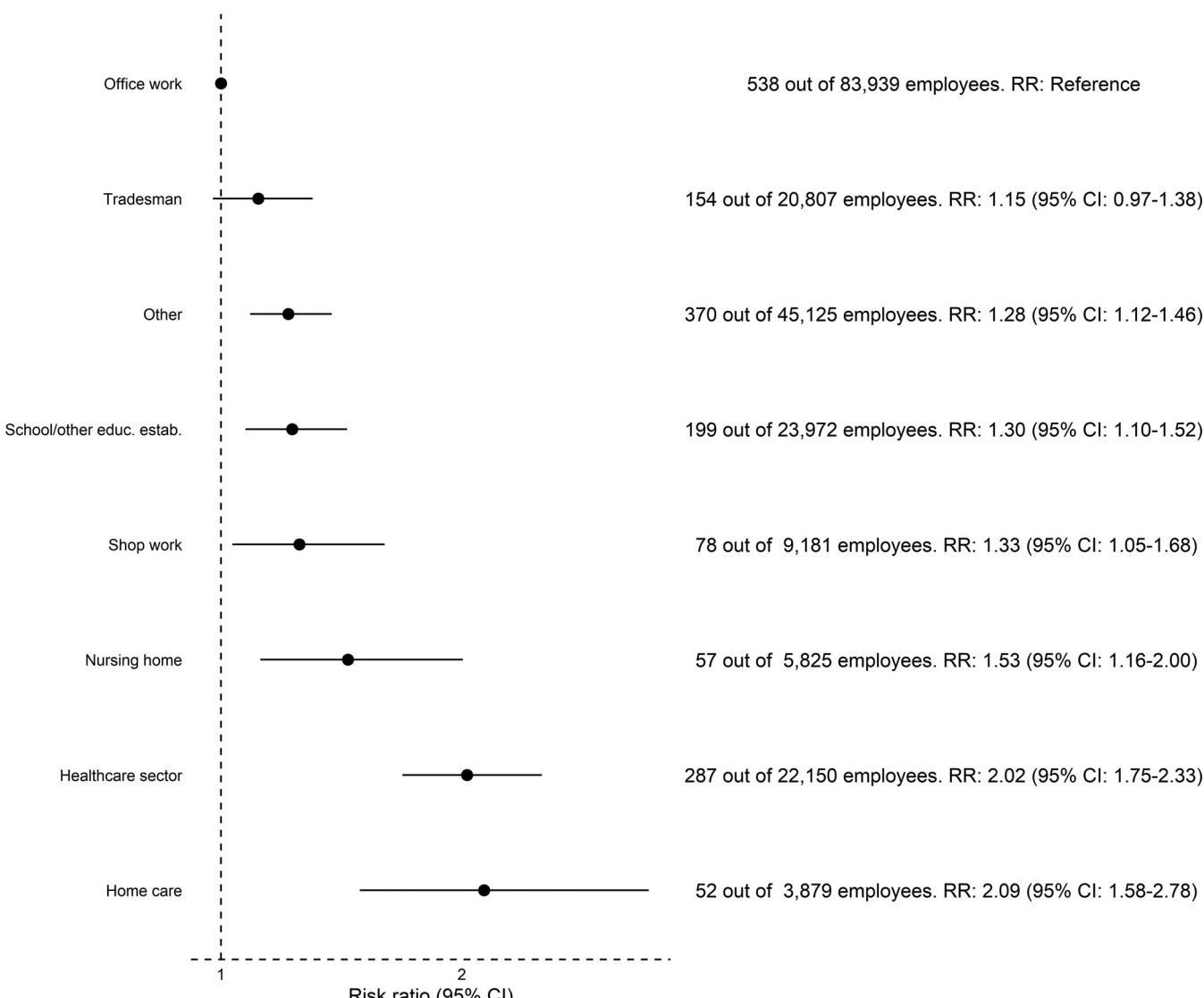

Office work — 538 out of 83,939 employees. RR: Reference

Tradesman — 154 out of 20,807 employees. RR: 1.15 (95% CI: 0.97-1.38)

Other — 370 out of 45,125 employees. RR: 1.28 (95% CI: 1.12-1.46)

School/other educ. estab. — 199 out of 23,972 employees. RR: 1.30 (95% CI: 1.10-1.52)

Shop work — 78 out of 9,181 employees. RR: 1.33 (95% CI: 1.05-1.68)

Nursing home — 57 out of 5,825 employees. RR: 1.53 (95% CI: 1.16-2.00)

Healthcare sector — 287 out of 22,150 employees. RR: 2.02 (95% CI: 1.75-2.33)

Home care — 52 out of 3,879 employees. RR: 2.09 (95% CI: 1.58-2.78)

Risk ratio (95% CI)

**FIG 4** Risk ratio for seropositivity in a subset of 193,646 working (full-time, part-time, or self-employed) participants. Participants in each profession were compared to participants in office work.

who have retired, which could contribute to our observations. However, in a recent report by the HOPE project, elderly people in Denmark were not found to report higher levels of self-quarantine when experiencing symptoms or when testing positive by PCR compared to younger people (23). During the study period, it was only possible to have PCR tests performed at hospitals or test facilities in the major cities. It may thus have been more difficult for older participants to be tested. Test facilities increased in Denmark during autumn/winter 2020 (19).

Ageusia and anosmia had the strongest correlation to seropositivity, consistent with previous findings (9, 10, 12). In general, we found that seropositive participants more frequently recalled having had symptoms compared to seronegative participants.

When stratifying for age groups, elderly participants reported symptoms less frequently. Selection of the healthiest elderly participants or comorbidities with associated symptoms and a long recall period may underestimate symptoms caused by SARS-CoV-2 infection. Our results are surprising because aging itself has been associated with more severe COVID-19 symptoms due to increased comorbidities with age and more aggressive clinical behavior (24). Nevertheless, the estimate of antibodies (comparable levels of IgG and IgM) was highest among elderly participants although they reported fewer symptoms

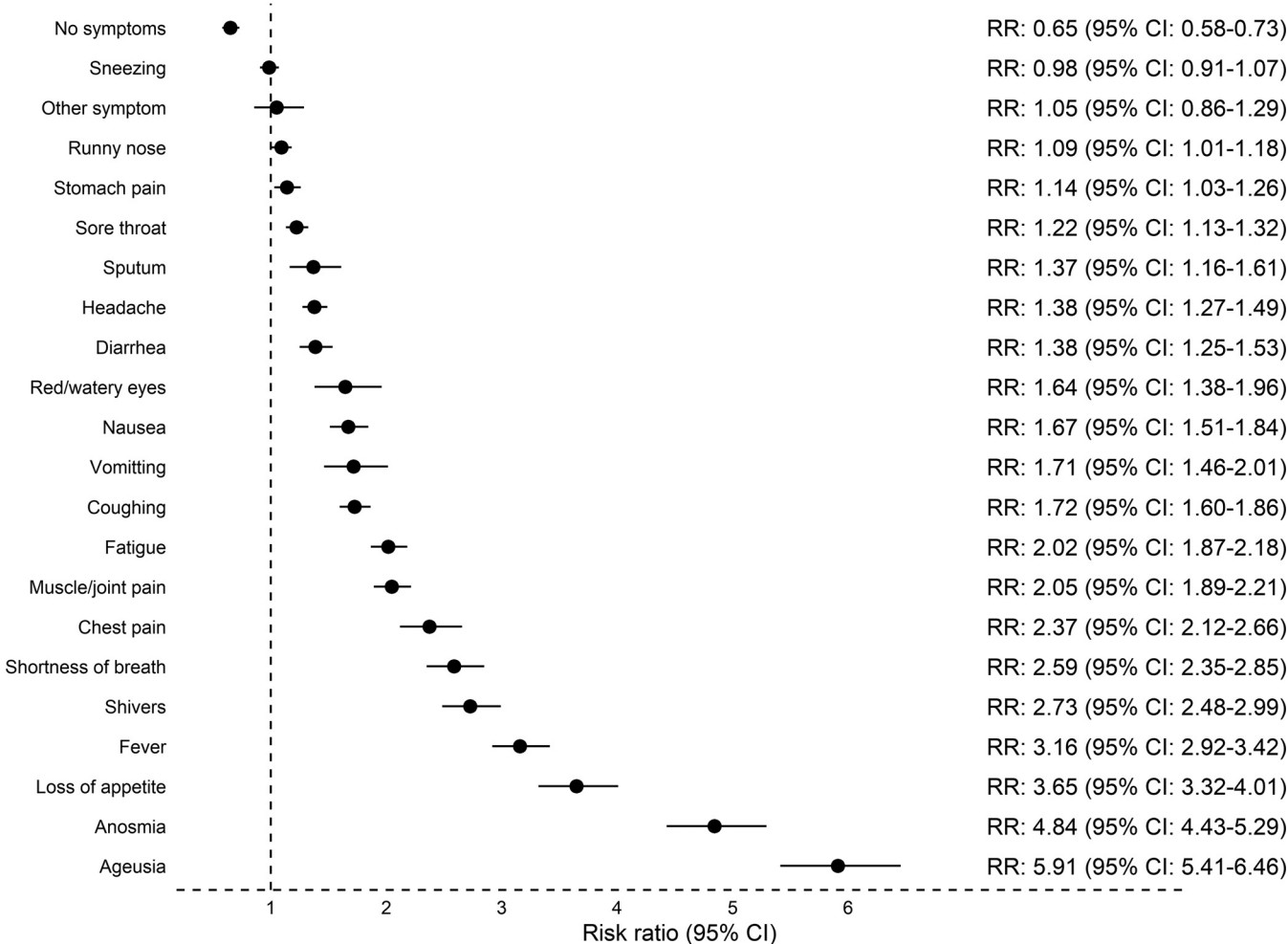

**FIG 5** Risk of seropositivity for individual symptoms. Analysis included 318,552 participants.

and had fewer tests. This had not been seen in other Danish studies, where seroprevalence was highest among the younger age groups (5, 8), except in a study of social housing areas where the seroprevalence was found increasing with age (12). As such, elderly participants may more often be subject to asymptomatic infections, thereby constituting an important subgroup that may warrant further attention.

**Occupation.** As previously reported, working in the health care sector was associated with a higher risk of seropositivity (9, 25). Working in home care or at nursing homes also increased the risk, as it involves working with patients and being in close physical contact to other persons (25). The proportion of females working in the health care sector is typically higher than males (26), possibly explaining the higher proportion of seropositive females. Conversely, participants with office jobs, and possibly better opportunities to work from home, have been at less risk of infection during the first infectious wave.

**Behavior and household.** We observed a high proportion of participants following the authority's recommendations to reduce the risk of SARS-CoV-2 infection. Remarkably, seropositive participants were slightly more compliant with these recommendations compared to seronegative participants on almost all the preventive measures. However, participants who are more attentive to recommendations, e.g., health care professionals could be more exposed to SARS-CoV-2 infection. As such, the effect of the authority's recommendations could be underestimated. Household composition is an important venue for transmission of infection due to household size and living conditions (27). Sustained close contact and crowded indoor environments pose a higher risk of transmission (28, 29). A metanalysis by Madewell et al. indicated that household and family members are at higher risk of infection compared with other

types of close contacts, and spouses were at higher risk compared with family contacts. Household crowding (e.g., number of people per room) may be more important for transmission than the total number of people per household (27). Our results showed seropositivity to be highest among smaller households with only two household members, possibly due to two person households often comprising couples with close contact, and thereby increased risk of transmission. This finding is also consistent with a previous preprint study on SARS-CoV-2 transmission within Danish households, which demonstrated a transmission pattern that was exponentially decreasing with the number of members in the household (20).

**Strengths and limitations.** This study had a broad national participation with 22% of the population invited and a response rate of 36.5% among the invitees for the questionnaire and 24.5% for the POCT. To determine the distribution of infectious disease, serological surveys with a representative sample of the wider population are important, particularly in the presence of asymptomatic individuals or incomplete ascertainment of those with symptoms.

This study has limitations. Recruitment by e-Boks might exclude persons without or limited access to this digital governmental information system and less technology-proficient individuals, or marginalized groups who have a higher risk of infection (11, 12). Also, e-Boks is only available for persons over the age of 15 years. A smaller proportion of residents may not have been able to read and understand Danish, English or Arabic. The recall period of symptoms was up to 7 months. Persons with a previous positive PCR may have been less inclined to participate, thereby resulting in selection bias and potentially underestimating the true seroprevalence. Conversely, particularly persons working in health care or nursing may have had an increased interest to know about possible protective immune status due to their working tasks and knowledge of former infection and/or increased exposure. The low seroprevalence at 0.79% in our study is a clear limitation and may be due to a low sensitivity of the POCT used or problems performing or reading the test results. The participant rate of 22% could be due to the requirements of online access, as invitation for participation was sent by e-Boks, an online platform, where participants sign up, answer questionnaires and report POCT results. This could probably have been optimized if not only online access was required for participation and if project staff were able to help with the performance either by visiting people at home or in their local community. The complexity of participants performing and reading the POCT could also have been a factor in the participation rate, however, with a total number of 318,522 participants. The sensitivity of the POCT used was lower than anticipated. The major importance of the study was the identification of several factors and risk associations nationwide and across multiple subgroups.

**Conclusion.** The seroprevalence result was somewhat hampered by a lower than expected performance of the POCT. This is likely due to a low sensitivity of the POCT or challenges relating to the reading of test results, and the main findings therefore relate to risk associations. We found that more emphasis from national and local authorities toward the risk of infection should be placed on age of tested individuals, type of occupation, as well as exposure in local communities and households.

## MATERIALS AND METHODS

**Study design and participation.** 1.3 million Danish citizens over the age of 15 years (22% of the population) were randomly drawn from the Civil Registration System and invited to participate via the governmental, personal, password-protected digital mailbox system (e-Boks) from September 25, 2020. e-Boks is linked to each individual's Danish personal registration number from age 15 years and above. Written information about the project was available in three different languages: Danish, English, and Arabic.

Participants were invited to complete a web-based questionnaire by a link (Enalyzer, Copenhagen, Denmark) in the invitation letter. The questionnaire included demographics, history of symptoms compatible with COVID-19, comorbidities, and substance use (see Supplemental text). In the questionnaire participants could further indicate if they wanted a point-of care rapid test (POCT) sent to their home address.

During October 2020 POCT testing for SARS-CoV-2 IgG and IgM antibodies was performed by participants. Answers to the questionnaire and POCT results were registered by the participant in a secondary separate questionnaire sent to their e-Boks and returned to Enalyzer.

Detailed information about the test-procedure was provided with the invitation and at the project website (www.vitesterdanmark.dk), including instructional video on how to perform the POCT, as well as videos with experts explaining aspects of the study. Social media (Facebook and Instagram) were used

for visualization. A call-center was set up for the participants for questions about the project, the questionnaire or how to perform the POCT.

Information about previous positive PCR test results among study participants was obtained from The Danish Microbiological Database, that has complete coverage of all microbiological samples analyzed by public laboratories (30).

**Detection of SARS-CoV-2 antibodies.** The Livzon POCT (Livzon Diagnostics, Zhuhai, Guangdong, China) was used. In spring 2020, Danish Regions (an organization for the five Danish regions) ordered Livzon POCT to be used in research in immunity and for prevalence studies of SARS-CoV-2 in Denmark. The POCT is a lateral flow chromatographic immunoassay rapid test for qualitative detection and differentiation of anti-SARS-CoV-2 IgG and IgM antibodies in whole blood, which yields results in 15 min. The manufacturer reported a combined test sensitivity (either IgG or IgM positive) of 90.6% (95% CI: 86.6%–93.4%) and a combined specificity (neither IgG nor IgM is positive) of 99.2% (95% CI: 97.6%–99.7%) (31). An in-house validation (cases = 150 individuals, controls = 600 individuals) showed sensitivity of 93.3% and 92.7% and specificity of 98.2% and 97.5% for each of the two batches, respectively (see Table S1). The case panel samples were obtained from convalescent individuals within 2 months of disease onset. Sensitivity and specificity by self-use has not previously been studied.

The POCT was sent out with a small container of isotonic saline, capillary tubes, and fingerprickers. Participants were instructed to add blood by fingerprick and isotonic saline to each of the two test cassettes (IgG and IgM). The test results were read by participants individually. Participants were considered positive when both control line and test line appeared, and inconclusive when no control line appeared or if the reading chamber was discolored by blood. Inconclusive test results were treated as negative. The test could not be repeated, as participants only received one POCT for both IgG and IgM.

**Outcome measures.** The primary outcome was to explore the association between SARS-CoV-2 infection, defined as a positive SARS-CoV-2 antibody self-test result (IgG and/or IgM), and putative risk factors for seropositivity.

The proportion of the study population with a positive antibody test for SARS-CoV-2 (as a proxy for previous infection) was a secondary outcome.

**Approvals, ethics, and registrations.** This study was performed as a national surveillance study under the authority task of the national infectious disease control institute Statens Serum Institut (SSI), Copenhagen, Denmark. According to Danish law national surveillance activities from SSI do not require any individual approval from an ethics committee. The study was performed in agreement with the Helsinki II declaration and registered with the Danish Data Protection Authorities (P-2020-901). Participation was voluntary and all data were self-reported. All personal data obtained in Enalyzer was kept in accordance with the general data protection regulation and data protection law stated by the Danish Data Protection Agency. Invitees received information about their legal rights and the use of their data in the invitation letter.

**Statistical analyses.** Participants were seropositive if they tested positive for IgG, IgM, or both antibodies. Baseline characteristics of seropositive compared to seronegative persons are presented as numbers and percentages for categorical values. Continuous values are presented as medians and interquartile ranges. The Wilcoxon rank test and chi-square test were used for comparisons of groups for continuous and categorial values. Unadjusted risk ratios (RR) with 95% confidence intervals (CI) were calculated for risk factors of seropositivity. We used logistic regression to calculate odds ratios (OR) for seropositivity with 95% CI adjusted for sex, age, and household size for participants exposed to COVID-19 infected patients within the household. Data on population and areal by municipality for 2020 was obtained from Statistics Denmark (32). For participants with previous positive PCR test, we calculated the proportion of seropositive participants. Further, for seropositive participants with an available date of POCT and PCR test, we investigated the proportion of seropositive participants according to time since PCR test. We analyzed the seroprevalence according to self-assed risk of being infected with SARS-CoV-2. Demographics were compared for responders and nonresponders to the questionnaire and compared for participants who provided the POCT results. $P < 0.05$ was considered statistically significant. Data management, statistical analyses, and figures were performed and created using R version 3.2.1.

## SUPPLEMENTAL MATERIAL

Supplemental material is available online only.

**SUPPLEMENTAL FILE 1**, PDF file, 1.1 MB.

## ACKNOWLEDGMENTS

We thank the Danish Ministry of Health (Grant 2012461), the Danish Patient Safety Authority, the Local Government Denmark, Danish Regions, Danish Patients, DaneAge Association, the Danish Medical Association, the Danish Nurses Organization, the Danish Heart Association, the Danish Cancer Society, the Danish Lung Association, the Danish National Organization for homeless people (SAND), the Danish Family Planning Association, and the Council for Ethnic Minorities for support of the study. The funders did not influence study design, conduct, or reporting.

The study was designed and initiated by K.F., B.S., R.S., H.U., and K.I. Data analysis was done by J.S. and K.F. The first draft was written by K.F., J.S., H.B., R.S., and K.I. All

authors have critically revised the manuscript and agree to be accountable for all aspects of the work. All authors approved the final version of the manuscript.

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
