## [Reviewer comments · Microbiology Spectrum]

Microbiology Spectrum

Testing Denmark: A Danish nationwide surveillance study of COVID-19

Kamille Fogh, Jarl Strange, Bibi Scharff, Alexandra Eriksen, Rasmus Hasselbalch, Henning Bundgaard, Susanne Nielsen, Charlotte Jørgensen, Christian Erikstrup, Jakob Norsk, Pernille Nielsen, Jonas Henrik Kristensen, Lars Østergaard, Svend Ellermann-Eriksen, Berit Andersen, Henrik Nielsen, Isik Johansen, Lothar Wiese, Lone Simonsen, Thea Fischer, Fredrik Folke, Freddy Lippert, Sisse Ostrowski, Thomas Benfield, Kåre Mølbak, Steen Ethelberg, Anders Koch, Ute Sönksen, Anne-Marie Vangsted, Tyra Krause, Anders Formsgaard, Henrik Ullum, Robert Skov, and Kasper Iversen

Corresponding Author(s): Kamille Fogh, Herlev and Gentofte Hospital, University of Copenhagen

Review Timeline:

Submission Date:	August 19, 2021
Editorial Decision:	September 27, 2021
Revision Received:	November 11, 2021
Accepted:	November 12, 2021

Editor: Miguel Martinez

Reviewer(s): The reviewers have opted to remain anonymous.

Transaction Report:

DOI: <https://doi.org/10.1128/Spectrum.01330-21>

September 27, 2021

Dr. Kamille Fogh
Herlev and Gentofte Hospital, University of Copenhagen
Dept. of Cardiology & Dept. of Emergency Medicine
Herlev
Denmark

Re: Spectrum01330-21 (Testing Denmark: A Danish nationwide surveillance study of COVID-19)

Dear Dr. Kamille Fogh:

Your manuscript has been considered by one reviewer recruited for their expertise in the field and myself. As the reviewer comments indicate, we felt that your manuscript contained interesting observations but that it required modification (see below) before it could be considered acceptable for publication.

Thank you for submitting your manuscript to Microbiology Spectrum. When submitting the revised version of your paper, please provide (1) point-by-point responses to the issues raised by the reviewers as file type "Response to Reviewers," not in your cover letter, and (2) a PDF file that indicates the changes from the original submission (by highlighting or underlining the changes) as file type "Marked Up Manuscript - For Review Only". Please use this link to submit your revised manuscript - we strongly recommend that you submit your paper within the next 60 days or reach out to me. Detailed information on submitting your revised paper are below.

Link Not Available

Sincerely,

Miguel Martinez

Journals Department
Reviewer comments:

Reviewer #2 (Comments for the Author):

Dear authors

Comments

I appreciate the opportunity to review the manuscript titled "Testing Denmark: A Danish nationwide surveillance study of COVID-19" by Fogh et al. It is an impressive national large-scale epidemiological surveillance study of SARS-CoV-2 in the Danish population with high number of participants, a well-developed and informative project webpage with instruction, information etc. The manuscript is well written and presented and has some great strengths such as the nationwide nature and the high participation with 318,552 individuals, corresponding to 22% of the population. However, the strengths do not out way my main concern which is that only 1/4 of the invited participated lowering the representativeness of the study and increasing the likelihood of bias. Another concerning issue is that only 29% of those with a positive PCR test were seropositive in the Point-of Care rapid Test used which is highly concerning as many studies show that people infected with COVID-19 develop antibodies

which remain detectable for several months afterward - some studies show up to 12 months. The likelihood of false positive in the PCR test is much lower compared with the risk of false negative (e.g. stated in ref. 3), i.e., that the discrepancy between PCR test and POCT is most likely due to the POCT performance. Thus, the validity of the seroprevalence assessment is questionable. The authors are aware of these issues and address them to some extent in the discussion but there are some limitations that still need to be addressed or elaborated on in my opinion, e.g., the discussion does not really discuss the problem with only 29% of confirmed PCR cases have positive POCT and the literature regarding persistence of antibodies over time.

The primary outcome is to explore the association between SARS-CoV-2 infection, defined as a positive SARS-CoV-2 antibody self-test result (IgG and/or IgM), and putative risk factors for seropositivity. But when the test most likely does not perform good, will the association then be valid if many are false negative e.g.? In the discussion (l. 298-230) the authors argue that seropositivity was low among participants who did not have a previous positive PCR test, indicating a high specificity of the POCT. However, one could argue that this also is the case with those who had a positive PCR test as one would expect much higher percentage than 29% to have antibodies. I would welcome more discussion of this limitation of the test used. Further, if the seronegative group include many false negative, one could question the observed associations. This also needs to be discussed more.

Please find more specific points below:

Importance

l. 29-30. This statement 'We found that more emphasis should be placed on occupation, exposure in local communities as well as age of participants' is rather overall. More emphasis in which way, how and who? Please elaborate (same in the conclusion)

l. 31. The authors state that this setup can be used as a model. Would the authors recommend to use POCT and would recommend the same approach having in mind that participation rate? I suggest rephrasing this or elaborate

Introduction

l. 48. I know it is included in the discussion, but I suggest adding seroprevalence estimates from the Danish studies in the introduction as this would be informative for the reader to know beforehand.

l. 50 This sentence does seem out of context. Please include more information about Test Denmark in the introduction and reference if possible.

l. 53. The aim was to determine the distribution of SARS-CoV-2 antibodies among Danish citizens. What do you mean by distribution? Do you refer to estimate of seropositivity/seroprevalence? In the discussion, you use the word seroprevalence and seropositivity. I suggest using other word that distribution which can be confusing.

Method

l. 58. What was the rationale behind restricting over the age of 15? Why not include children? Do all from age 15 years and older have e-Boks?

l. 65. Write POCT out first time mentioned in the manuscript

l. 78. Why or what was the rationale behind the decision to use Livzon POCT to assess the antibody levels in this study?

Results

l. 139. There is a higher proportion that is positive for IgM than IgG? Would one not expect that IgG was more present as it is longer lasting?

l.139. I suggest including confidence interval to the crude prevalence and include test adjusted prevalence.

l. 146. Was the difference in seroprevalence between male and females significant?

l. 148-149. What is meant by numerically higher proportion? And is this compared with the other women or men? Are these observations statistically significant? Maybe add this information in suppl figure 4?

l. 151. In table 1, you report ever smokers but in supplementary figure 4, you report never, more than 5 years ago, within 5 years and occasionally? Why not also include ever smoker in suppl fig 4 when this is what you present in table 1? NA in panel B (suppl fig 4) with weekly alcohol...abbreviation should be included as footnote or in the title. And when do you use NA - does that encompass teetotaller or? Panel C, define how you divide BMI into groups

l. 160. Table 3 is mentioned before table 2. In table 3, the variable MISSING with n=693, 331 and 1024. What is missing? What does that encompass?

l. 162. No clear association was found between seropositivity and population density but was there difference in seropositivity according to geography? Suppl fig 8 shows some green and some red areas.

Reg. suppl fig 9, as there was no clear association and limited mentioning of this aspect in the manuscript, I suggest omitting this figure

l. 177. I suggest adding information about work, exposure to infected, PCR testing to the baseline characteristics in table 1.

l. 184. Supplementary table 11. Are these symptoms among all participants or only those who had positive test? Please make more clear

Discussion

I. 202-I. 217. The discussion does not really discuss the problem with only 29% of confirmed PCR cases have positive POCT. This discrepancy has to be discussed. The antibody response of IgM and IgG is found to be highest about 2-4 weeks after symptom onset and decrease afterwards. The authors argue that for seronegative, longer time had passed from a previously positive PCR test than for seropositive. However, this statement has to be discussed in relation to relevant literature, e.g., regarding the persistence of antibodies over time. I find that it seems unlikely according to the scientific literature that 70% of those with positive PCR test do not present with antibodies. Thus, the antibody test used (POCT) is a rather big limitation in this otherwise impressive study and needs more attention in the discussion.

And would you expect that IgM was more prevalent than IgG over time like found in this study? Please discuss this finding

I. 207. You state: 'The diagnostic testing window is also of importance as the study was performed seven to eight months after the first COVID-19 case in Denmark.'. But how do you expect this to influence your results? Please elaborate

I. 216. I might have missed this information but where is the data regarding seropositivity among participants who did not previously have a previous positive PCR test? One could also argue that this was true among those with a positive PCR test as one would expect this number to be much higher.

L. 225. Please discuss in connection with the other Danish studies why your estimate is lower both the population based and the convenient sample study.

I. 231. Sending our test material to participants at home, may enabled inclusion of vulnerable and elderly susceptible to infection who otherwise would not have participated. But what about those, e.g., maybe elderly, who may have difficulties to perform the test at home. They may e.g. not be as liable to access the demonstration online. Could this be one reason for the lower participation among the elderly? Please include in the discussion

I. 240. Why do elderly test less frequently? Does e.g., distance to testing setting play a role?

I. 253. Have others reported higher antibody levels among elderly? What do you mean by distribution of antibodies? Literature regarding elderly and symptoms need to be addresses in connection with this notion that 'As such, elderly participants may more often be subject to asymptomatic infections, thereby constituting an important subgroup that may warrant further attention'.

I. 256 to 258. This sentence regarding working age and attending PCR testing - is that in relation to the section I. 240-245. If yes, please move the section above. If not, please make more clear

I. 287. Yes, serological surveys with a representative sample of the wider population are important. But is this the case in this study? Please argue for this statement

I. 293. What about the difficulty to perform the antibody test? It should be added as a potential source of bias

I. 298. I think that the antibody test used must be discussed as a limitation along with the fact that only 29% of those with positive PCR had positive POCT. Further, there is a need to address the low participation rate and representativeness of the sample.

I. 305. This statement is rather overall. More emphasis in which way?

Table 1.

Male (%) 113,412 (422) - should be 42.2% I assume.

Fig 3. Mentioning subset in the title, you mean subset based on the categories on the Y-axis?

Suppl fig 7. Why is n=804?

Staff Comments:

Preparing Revision Guidelines

Please return the manuscript within 60 days; if you cannot complete the modification within this time period, please contact me. If you do not wish to modify the manuscript and prefer to submit it to another journal, please notify me of your decision immediately so that the manuscript may be formally withdrawn from consideration by Microbiology Spectrum.

Dear authors

Comments

I appreciate the opportunity to review the manuscript titled "Testing Denmark: A Danish nationwide surveillance study of COVID-19" by Fogh et al. It is an impressive national large-scale epidemiological surveillance study of SARS-CoV-2 in the Danish population with high number of participants, a well-developed and informative project webpage with instruction, information etc. The manuscript is well written and presented and has some great strengths such as the nationwide nature and the high participation with 318,552 individuals, corresponding to 22% of the population. However, the strengths do not out way my main concern which is that only ¼ of the invited participated lowering the representativeness of the study and increasing the likelihood of bias. Another concerning issue is that only 29% of those with a positive PCR test were seropositive in the Point-of Care rapid Test used which is highly concerning as many studies show that people infected with COVID-19 develop antibodies which remain detectable for several months afterward – some studies show up to 12 months. The likelihood of false positive in the PCR test is much lower compared with the risk of false negative (e.g. stated in ref. 3), i.e., that the discrepancy between PCR test an POCT is most likely due to the POCT performance. Thus, the validity of the seroprevalence assessment is questionable. The authors are aware of these issues and address them to some extend in the discussion but there are some limitations that still need to be addressed or elaborated on in my opinion, e.g., the discussion does not really discuss the problem with only 29% of confirmed PCR cases have positive POCT and the literature regarding persistence of antibodies over time.

The primary outcome is to explore the association between SARS-CoV-2 infection, defined as a positive SARS-CoV-2 antibody self-test result (IgG and/or IgM), and putative risk factors for seropositivity. But when the test most likely does not perform good, will the association then be valid if many are false negative e.g.? In the discussion (l. 298-230) the authors argue that seropositivity was low among participants who did not have a previous positive PCR test, indicating a high specificity of the POCT. However, one could argue that this also is the case with those who had a positive PCR test as one would expect much higher percentage than 29% to have antibodies. I would welcome more discussion of this limitation of the test used. Further, if the seronegative group include many false negative, one could question the observed associations. This also needs to be discussed more.

Please find more specific points below:

Importance

I. 29-30. This statement 'We found that more emphasis should be placed on occupation, exposure in local communities as well as age of participants' is rather overall. More emphasis in which way, how and who? Please elaborate (same in the conclusion)

I. 31. The authors state that this setup can be used as a model. Would the authors recommend to use POCT and would recommend the same approach having in mind that participation rate? I suggest rephrasing this or elaborate

Introduction

I. 48. I know it is included in the discussion, but I suggest adding seroprevalence estimates from the Danish studies in the introduction as this would be informative for the reader to know beforehand.

I. 50 This sentence does seem out of context. Please include more information about Test Denmark in the introduction and reference if possible.

I. 53. The aim was to determine the distribution of SARS-CoV-2 antibodies among Danish citizens. What do you mean by distribution? Do you refer to estimate of seropositivity/seroprevalence? In the discussion, you use the word seroprevalence and seropositivity. I suggest using other word that distribution which can be confusing.

Method

I. 58. What was the rationale behind restricting over the age of 15? Why not include children? Do all from age 15 years and older have e-Boks?

I. 65. Write POCT out first time mentioned in the manuscript

I. 78. Why or what was the rationale behind the decision to use Livzon POCT to assess the antibody levels in this study?

Results

I. 139. There is a higher proportion that is positive for IgM than IgG? Would one not expect that IgG was more present as it is longer lasting?

I.139. I suggest including confidence interval to the crude prevalence and include test adjusted prevalence.

I. 146. Was the difference in seroprevalence between male and females significant?

I. 148-149. What is meant by numerically higher proportion? And is this compared with the other women or men? Are these observations statistically significant? Maybe add this information in suppl figure 4?

I. 151. In table 1, you report ever smokers but in supplementary figure 4, you report never, more than 5 years ago, within 5 years and occasionally? Why not also include ever smoker in suppl fig 4 when this is what you present in table 1? NA in panel B (suppl fig 4) with weekly alcohol...abbreviation should be included as footnote or in the title. And when do you use NA – does that encompass teetotaler or? Panel C, define how you divide BMI into groups

I. 160. Table 3 is mentioned before table 2. In table 3, the variable MISSING with n=693, 331 and 1024. What is missing? What does that encompass?

I. 162. No clear association was found between seropositivity and population density but was there difference in seropositivity according to geography? Suppl fig 8 shows some green and some red areas. Reg. suppl fig 9, as there was no clear association and limited mentioning of this aspect in the manuscript, I suggest omitting this figure

I. 177. I suggest adding information about work, exposure to infected, PCR testing to the baseline characteristics in table 1.

I. 184. Supplementary table 11. Are these symptoms among all participants or only those who had positive test? Please make more clear

Discussion

I. 202-I. 217. The discussion does not really discuss the problem with only 29% of confirmed PCR cases have positive POCT. This discrepancy has to be discussed. The antibody response of IgM and IgG is found to be highest about 2-4 weeks after symptom onset and decrease afterwards. The authors argue that for seronegative, longer time had passed from a previously positive PCR test than for seropositive. However, this statement has to be discussed in relation to relevant literature, e.g., regarding the persistence of antibodies over time. I find that it seems unlikely according to the scientific literature that 70% of those with positive PCR test do not present with antibodies. Thus, the antibody test used (POCT) is a rather big limitation in this otherwise impressive study and needs more attention in the discussion.

And would you expect that IgM was more prevalent than IgG over time like found in this study? Please discuss this finding

I. 207. You state: 'The diagnostic testing window is also of importance as the study was performed seven to eight months after the first COVID-19 case in Denmark.'. But how do you expect this to influence your results? Please elaborate

I. 216. I might have missed this information but where is the data regarding seropositivity among participants who did not previously have a previous positive PCR test? One could also argue that this was true among those with a positive PCR test as one would expect this number to be much higher.

L. 225. Please discuss in connection with the other Danish studies why your estimate is lower both the population based and the convenient sample study.

I. 231. Sending our test material to participants at home, may enabled inclusion of vulnerable and elderly susceptible to infection who otherwise would not have participated. But what about those, e.g., maybe elderly, who may have difficulties to perform the test at home. They may e.g. not be as liable to access the demonstration online. Could this be one reason for the lower participation among the elderly? Please include in the discussion

I. 240. Why do elderly test less frequently? Does e.g., distance to testing setting play a role?

I. 253. Have others reported higher antibody levels among elderly? What do you mean by distribution of antibodies? Literature regarding elderly and symptoms need to be addresses in connection with this notion that 'As such, elderly participants may more often be subject to asymptomatic infections, thereby constituting an important subgroup that may warrant further attention'.

I. 256 to 258. This sentence regarding working age and attending PCR testing – is that in relation to the section I. 240-245. If yes, please move the section above. If not, please make more clear

I. 287. Yes, serological surveys with a representative sample of the wider population are important. But is this the case in this study? Please argue for this statement

I. 293. What about the difficulty to perform the antibody test? It should be added as a potential source of bias

I. 298. I think that the antibody test used must be discussed as a limitation along with the fact that only 29% of those with positive PCR had positive POCT. Further, there is a need to address the low participation rate and representativeness of the sample.

I. 305. This statement is rather overall. More emphasis in which way?

Table 1.

Male (%) 113,412 (422) – should be 42.2% I assume.

Fig 3. Mentioning subset in the title, you mean subset based on the categories on the Y-axis?

Suppl fig 7. Why is n=804?

8. November 2021

To Editor

Miguel Martinez

Microbiology Spectrum

Dear Miguel Martinez

Thank you for the opportunity to revise and improve our manuscript

“Testing Denmark: A Danish nationwide surveillance study of COVID-19”.

We appreciate the time and effort that the reviewers have spent on the revision of our manuscript, and we have carefully addressed all comments and concerns from the reviewers in the point-by-point reply below.

We consider the paper much improved and hope that you will consider it for publication in *The Microbiology Spectrum*.

Sincerely yours

Kamille Fogh, MD, Ph.d. student
Department of Cardiology and Department of Emergency Medicine
Herlev-Gentofte Hospital
Borgmester Ib Juuls Vej 1
DK - 2730 Herlev
T +45 2679 8310
E kamille.fogh.01@regionh.dk

Reviewer #2:

Placement of revision in each response refers to placement in clean version (not with track changes).

Importance

1. I. 29-30. This statement 'We found that more emphasis should be placed on occupation, exposure in local communities as well as age of participants' is rather overall. More emphasis in which way, how and who? Please elaborate (same in the conclusion)

Response to Reviewer, comment 1:

Thank you for this suggestion. We have changed the abstract.

Prior to revision:

We found that more emphasis should be placed on occupation, exposure in local communities as well as age of participants.

Revision (page 4, l. 29-31):

We found that more emphasis from national and local authorities towards the risk of infection should be placed on age of tested individuals, type of occupation, as well as exposure in local communities and households.

2. I. 31. The authors state that this setup can be used as a model. Would the authors recommend to use POCT and would recommend the same approach having in mind that participation rate? I suggest rephrasing this or elaborate

Response to Reviewer, comment 2:

Thank you for the question. Sending out POCT to participants' home addresses was a way to include a broad proportion of the population, including both vulnerable and older participants. The participation rate of 22%, could have been higher. This may be due to problems performing the POCT at home. However, we got a demographically representative sample size for data analysis. With this setup it is possible to include a broad sample of the population, including both healthy and vulnerable individuals, as self-testing at home is convenient. Sending POCT to participants' homes can be used as a supplementary model in future general test strategy for ongoing monitoring, but improvements in instructional material with the possibility of repeating the POCT if it failed or was inconclusive would be beneficial to decrease the frequency of inconclusive results. Furthermore, it is necessary to keep in mind the presumed reduced sensitivity, which is at least partially counterbalanced by large sample sizes.

Prior to revision:

Nationwide information can be difficult to gather and the study design in question presents a novel way for conducting future studies. Additionally, this setup can be used as a model for ongoing monitoring of COVID-19 immunity in the population, both from past infection and from vaccination against SARS-CoV-2.

Revision (page 4, l. 31-36):

To meet the challenge that broad nationwide information can be difficult to gather. This study design sets the stage for a novel way of conducting studies. Additionally, this study design can be used as a supplementary model in future general test strategy for ongoing monitoring of COVID-19 immunity in the population, both from past infection and from vaccination against SARS-CoV-2, however, with attention to the complexity of performing and reading the POCT at home.

Introduction

3. I. 48. I know it is included in the discussion, but I suggest adding seroprevalence estimates from the Danish studies in the introduction as this would be informative for the reader to know beforehand.

Response to Reviewer, comment 3:

We agree with the reviewer and have made the following revision:

Prior to revision:

The seroprevalence has been reported for selections of the Danish population (5, 8-12) but hitherto no national investigation of this scale has been performed in Denmark.

Revision (page 5, l. 51-53):

The seroprevalence has been reported for selections of the Danish population in summer and autumn 2020 with estimates of seroprevalence of approximately 2.0% (5, 8-12) but hitherto no national investigation at this scale has been performed in Denmark.

4. I. 50 This sentence does seem out of context. Please include more information about Test Denmark in the introduction and reference if possible.

Response to Reviewer, comment 4:

Thank you for the comment. "Testing Denmark" is the name of the study which will be mentioned in the method section.

Prior to revision:

"Testing Denmark" was a nationwide surveillance study of SARS-CoV-2 infection in the Danish population, launched in September 2020.

Revision (page 5, l.54-55):

This study "Testing Denmark" was a nationwide surveillance study of SARS-CoV-2 infection in the Danish population, launched in September 2020.

5. L. 53. The aim was to determine the distribution of SARS-CoV-2 antibodies among Danish citizens. What do you mean by distribution? Do you refer to estimate of seropositivity/seroprevalence? In the discussion, you use the word seroprevalence and seropositivity. I suggest using other word that distribution which can be confusing.

Response to Reviewer, comment 5:

Thank you for this suggestion. We apologize for the confusion.

Prior to revision:

The aim of this study was to explore possible risk factors for seropositivity by questionnaire data, and to determine the distribution of SARS-CoV-2 antibodies among Danish citizens.

Revision (page 5, l. 56-57):

The aim of this study was to explore possible risk factors for seropositivity by questionnaire data and to estimate the seroprevalence of SARS-CoV-2 antibodies among Danish citizens.

Method

6. I. 58. What was the rationale behind restricting over the age of 15? Why not include children? Do all from age 15 years and older have e-Boks?

Response to Reviewer, comment 6:

The governmental, personal, password-protected digital mailbox system (e-Boks) was used as it is a feasible way to invite as many participants as possible and covers the Danish population. However, e-Boks is only available for persons over the age of 15 years. This is a limitation of the study and has also been added to the limitations section.

Prior to revision:

1.3 million Danish citizens over the age of 15 years (22 % of the population) were randomly drawn from the Civil Registration System and invited to participate via the governmental, personal, password-protected digital mailbox system (e-Boks) from September 25, 2020.

Revision (page 6, l.61-65):

1.3 million Danish citizens over the age of 15 years (22 % of the population) were randomly drawn from the Civil Registration System and invited to participate via the governmental, personal, password-protected digital mailbox system (e-Boks) from September 25, 2020. e-Boks is linked to each individuals' Danish personal registration number from the age 15 years and above.

Added to the revised manuscript (page 16, l. 310)

Also, e-Boks is only available for persons over the age of 15.

7. I. 65. Write POCT out first time mentioned in the manuscript

Response to Reviewer, comment 7:

We agree and have added this.

Prior to revision:

During October 2020 the POCT testing for SARS-CoV-2 IgG and IgM antibodies was performed by participants.

Revision (page 6, l.70-71):

During October 2020 the point-of-care rapid test (POCT) testing for SARS-CoV-2 IgG and IgM antibodies was performed by participants.

8. I. 78. Why or what was the rationale behind the decision to use Livzon POCT to assess the antibody levels in this study?

Response to Reviewer, comment 8:

Thank you for this comment. In the spring 2020, Livzon POCT was ordered from the Danish Regions to use in research in immunity and prevalence studies of SARS-CoV-2 in Denmark.

Prior to revision:

The Livzon POCT (Livzon Diagnostics, Zhuhai, Guangdong, China) was used.

Revision (page 7, l.83-85):

The Livzon POCT (Livzon Diagnostics, Zhuhai, Guangdong, China) was used. In spring 2020, Danish Regions (an organization for the five Danish regions) ordered Livzon POCT to be used in research in immunity and for prevalence studies of SARS-CoV-2 in Denmark.

Results

9. I. 139. There is a higher proportion that is positive for IgM than IgG? Would one not expect that IgG was more present as it is longer lasting?

Response to Reviewer, comment 9:

Thank you for this comment. We have added the answer below in the “Discussion” section.

Added to revised manuscript (p. 12, l.216-219):

From September 2020 the incidence of infected people in Denmark increased, peaking in December 2020. This could explain the higher proportion of IgM positive than IgG positive found in this study. The first infection wave in spring 2020 was minor, fewer were therefore infected back then, resulting in less with IgG antibodies and more with IgM antibodies during the study period (22).

Reference 22:

Hansen CH, Michlmayr D, Gubbels SM, Molbak K, Ethelberg S. Assessment of protection against reinfection with SARS-CoV-2 among 4 million PCR-tested individuals in Denmark in 2020: a population-level observational study. *Lancet*. 2021;397(10280):1204-12.

10. I.139. I suggest including confidence interval to the crude prevalence and include test adjusted prevalence.

Response to Reviewer, comment 10:

This is a relevant suggestion. In fact, we previously tried to estimate test adjusted prevalence using the Rogan and Gladen method (Rogan WJ et al) as done previously (Iversen K et al). However, due to the low seroprevalence and the sensitivity and specificity of the POCT, it was not possible to estimate test adjusted prevalence.

Reference:

Rogan WJ, Gladen B. Estimating prevalence from the results of a screening test. *Am J Epidemiol* 1978; 107: 71–76.

Iversen K, Bundgaard H, Hasselbalch RB, Kristensen JH, Nielsen PB, Pries-Heje M, et al. Risk of COVID-19 in health-care workers in Denmark: an observational cohort study. *Lancet Infect Dis*. 2020;20(12):1401-8.

11. I. 146. Was the difference in seroprevalence between male and females significant?

Response to Reviewer, comment 11:

The line in question refers to Table 1 which shows the seroprevalence according to sex with a corresponding p-value yielded from the chi-squared test. For males 113,412 (42.2%) were seronegative and 1,012 (40.2%) were seropositive with a p-value of 0.041 when compared with females. While the difference is statistically significant, the clinical significance is minor, and the p-value is likely to be a result of the large sample size. The difference is not attributable to sex per se, but more likely to do with demographic differences between sex (i.e. more females working in the healthcare sector which confers an increased risk for seropositivity). Moreover, the text in question refers to females but Table 1 lists males, potentially confusing the reader. As such, we have revised the sentence in question.

Prior to revision:

Women were more likely to be seropositive (Table 1 and Supplementary figure 3).

Revision (page 10, l.154-155):

The seroprevalence was statistically significantly lower for males. However, the clinical difference was minor (Table 1 and Supplementary Figure 3).

12. I. 148-149. What is meant by numerically higher proportion? And is this compared with the other women or men? Are these observations statistically significant? Maybe add this information in suppl figure 4?

Response to Reviewer, comment 12:

We agree with the reviewer that the sentence is ambiguous. The comparison is to be interpreted as among participants smoking > 10 cigarettes per day, the seroprevalence was numerically higher among females compared with males. However, in the manuscripts current form, no statistical test has been performed for these comparisons. To avoid any confusion, we have now performed chi-squared tests between groups and provided the p-values in the figure. For complete revision of the figures, please see "Response to Reviewer, comment 13".

13. I. 151. In table 1, you report ever smokers but in supplementary figure 4, you report never, more than 5 years ago, within 5 years and occasionally? Why not also include ever smoker in suppl fig 4 when this is what you present in table 1? NA in panel B (suppl fig 4) with weekly alcohol...abbreviation should be included as footnote or in the title. And when do you use NA - does that encompass teetotaller or? Panel C, define how you divide BMI into groups

Response to Reviewer, comment 13:

We thank the reviewer for these suggestions. Along with the revisions above, we have made the following revisions:

Regarding smoking groups:

The "Ever"-smoking group is now included in Supplementary Figure 4.

Prior to revision:

Revision:

Regarding alcohol groups:

The NA encompasses participants, who did not fill out weekly alcohol consumption in the questionnaire. As self-reported alcohol consumption is prone to underreporting, we included this group in the figure to see if there was a marked heterogeneity when compared to other groups. However, we acknowledge this was not accurately reflected by the figure. Thus, the group has been changed from “NA” to “Missing” further elaborated in the figure legend.

Prior to revision:

Revision:

Regarding BMI groups:

The cut-offs for BMI groups have been added to the figure legend.

Prior to revision:

Revision:

Revised figure legend:

Supplementary figure 4: SARS-CoV-2 seropositive % according to smoking habits, weekly alcohol consumption, and BMI stratified for sex. Red bar represents females, blue bar represents males. Numbers above bars represent number of participants in each group. P-values above bars represent chi-squared comparisons of males and females within groups. For panel B, the “Missing”-group encompasses participants who did not fill

out weekly alcohol consumption in the questionnaire. For panel C, underweight, normal weight, pre-obesity and obese corresponds to BMI < 18.5, > 18.5 to 25, > 25 to 30, > 30.

14. I. 160. Table 3 is mentioned before table 2. In table 3, the variable MISSING with n=693, 331 and 1024. What is missing? What does that encompass?

Response to Reviewer, comment 14:

Regarding Table 3 and Table 2:

We apologize for this mistake. We have corrected the order of the figures in the manuscript.

Regarding Table 3, missing variable:

This encompasses participants who did not have both and available date for positive PCR and POCT. As such, we were not able to calculate days between positive PCR and POCT. This also means that of the total 1,828 participants in Table 2, 1,024 had missing information on days between positive PCR and POCT. The 804 participants (1,828-1,024) with information are represented in Supplementary Figure 7.

Regarding Table 2:

We have revised the Table 2 legend to detail what the Missing-category encompasses.

Prior to revision:

Table 3: Characteristics of the study cohort who previously testes positive on PCR test.

Full cohort	Seronegative	Seropositive	Total	p
n	1,296	532	1,828	
Age (years) (median [IQR])	47 [31-59])	51 [40-61]	49 [34-59]	<0.001
Male (%)	480 (37.0)	233 (43.8)	713 (39.0)	0.008
Body mass index (median [IQR])	24.9 [22.4, 28.4]	25.6 [23.0, 29.1]	25.1 [22.6, 28.7]	0.003
Days between pos. PCR and POCT (median [IQR])	58 [26, 188]	38 [23, 176]	46.5 [25, 187]	0.082
Missing	693	331	1,024	
Comorbidities (%)				
Myocardial infarction	26 (2.0)	11 (2.1)	37 (2.0)	1.000
Stroke	31 (2.4)	17 (3.2)	48 (2.6)	0.415
Hypertension	257 (19.8)	129 (24.2)	386 (21.1)	0.041
Diabetes	67 (5.2)	38 (7.1)	105 (5.7)	0.124
Cancer	75 (5.8)	33 (6.2)	108 (5.9)	0.815
Rheumatoid arthritis	72 (5.6)	31 (5.8)	103 (5.6)	0.907
COPD	46 (3.5)	21 (3.9)	67 (3.7)	0.784
Asthma	202 (15.6)	84 (15.8)	286 (15.6)	0.970
Other chronic disease	211 (16.3)	84 (15.8)	295 (16.1)	0.850
Alcohol use* (%)	144 (12.5)	56 (11.7)	22 (12.3)	0.708
Ever smoker (%)	607 (46.8)	278 (52.3)	885 (48.4)	0.040

Revision:

Table 2: Characteristics of the study cohort who previously testes positive on PCR test.

Full cohort	Seronegative	Seropositive	Total	p
n	1,296	532	1,828	
Age (years) (median [IQR])	47 [31-59])	51 [40-61]	49 [34-59]	<0.001
Male (%)	480 (37.0)	233 (43.8)	713 (39.0)	0.008
Body mass index (median [IQR])	24.9 [22.4, 28.4]	25.6 [23.0, 29.1]	25.1 [22.6, 28.7]	0.003
Days between pos. PCR and POCT (median [IQR])	58 [26, 188]	38 [23, 176]	46.5 [25, 187]	0.082
Missing*	693	331	1,024	
Comorbidities (%)				
Myocardial infarction	26 (2.0)	11 (2.1)	37 (2.0)	1.000
Stroke	31 (2.4)	17 (3.2)	48 (2.6)	0.415
Hypertension	257 (19.8)	129 (24.2)	386 (21.1)	0.041
Diabetes	67 (5.2)	38 (7.1)	105 (5.7)	0.124
Cancer	75 (5.8)	33 (6.2)	108 (5.9)	0.815
Rheumatoid arthritis	72 (5.6)	31 (5.8)	103 (5.6)	0.907
COPD	46 (3.5)	21 (3.9)	67 (3.7)	0.784
Asthma	202 (15.6)	84 (15.8)	286 (15.6)	0.970
Other chronic disease	211 (16.3)	84 (15.8)	295 (16.1)	0.850
Alcohol use* (%)	144 (12.5)	56 (11.7)	22 (12.3)	0.708
Ever smoker (%)	607 (46.8)	278 (52.3)	885 (48.4)	0.040

*Missing encompasses participants who did not have an available date of both positive PCR and POCT. Thus, days between positive PCR and POCT could not be calculated for these participants.

15. I. 162. No clear association was found between seropositivity and population density but was there difference in seropositivity according to geography? Suppl fig 8 shows some green and some red areas. Reg. suppl fig 9, as there was no clear association and limited mentioning of this aspect in the manuscript, I suggest omitting this figure

Response to Reviewer, comment 15:

A good question. We preplanned analysis to investigate geographical variations in seroprevalence (Supplementary Figure 8) and variations in seroprevalence according to population density (Supplementary Figure 9). However, the seroprevalence was too low to accurately analyze geographical variations. We agree with Reviewer and have omitted Supplementary Figure 9.

16. I. 177. I suggest adding information about work, exposure to infected, PCR testing to the baseline characteristics in table 1.

Response to Reviewer, comment 16:

We appreciate the suggestion and the information has been added to Table 1.

Prior to revision:

Table 1: Baseline characteristics of the study cohort on sex, age, BMI, smoking, alcohol use, previous test result and comorbidities stratified by seropositivity.

Full cohort	Seronegative	Seropositive	p
n	316,033	2,519	
Age (years) (median [IQR])	53 [39-64])	55 [42-64]	0.041
Male (%)	113,412 (42.2)	1,012 (40.2)	<0.001
Body mass index (median [IQR])	25.4 [22.8, 28.7]	25.5 [23, 29]	0.115
Ever smoker (%)	168,024 (53.2)	1,375 (54.6)	0.161
Alcohol use* (%)	36,747 (12.9)	302 (13.5)	0.443
Comorbidities (%)			
Myocardial infarction	6562 (2.1)	59 (2.3)	0.389
Stroke	9067 (2.9)	91 (3.6)	0.030
Hypertension	82215 (26.0)	711 (28.2)	0.013
Diabetes	17528 (5.5)	165 (6.6)	0.032
Cancer	23250 (7.4)	185 (7.3)	1.000
Rheumatoid arthritis	19309 (6.1)	176 (7.0)	0.074
COPD	13872 (4.4)	150 (6.0)	<0.001
Asthma	43996 (13.9)	375 (14.9)	0.172
Other chronic disease	56134 (17.8)	456 (18.1)	0.675

*Alcohol use: Reporting >7 units of alcohol a week for females or >14 units of alcohol for males

Revision:

Table 1: Baseline characteristics of the study cohort on sex, age, BMI, smoking, alcohol use, previous test result and comorbidities stratified by seropositivity.

Full cohort	Seronegative	Seropositive	p
n	316,033	2,519	
Age (years) (median [IQR])	53 [39-64])	55 [42-64]	0.041
Male (%)	113,412 (42.2)	1,012 (40.2)	<0.001
Body mass index (median [IQR])	25.4 [22.8, 28.7]	25.5 [23, 29]	0.115
Ever smoker (%)	168,024 (53.2)	1,375 (54.6)	0.161
Alcohol use* (%)	36,747 (12.9)	302 (13.5)	0.443
Comorbidities (%)			
Myocardial infarction	6562 (2.1)	59 (2.3)	0.389
Stroke	9067 (2.9)	91 (3.6)	0.030
Hypertension	82215 (26.0)	711 (28.2)	0.013
Diabetes	17528 (5.5)	165 (6.6)	0.032
Cancer	23250 (7.4)	185 (7.3)	1.000
Rheumatoid arthritis	19309 (6.1)	176 (7.0)	0.074
COPD	13872 (4.4)	150 (6.0)	<0.001
Asthma	43996 (13.9)	375 (14.9)	0.172
Other chronic disease	56134 (17.8)	456 (18.1)	0.675
Work type			
Not working**	123,959 (39.2)	947 (37.6)	
Office work***	83,401 (43.4)	538 (34.2)	
Tradesman	20,653 (10.8)	154 (9.8)	

School/other educ. Stab.	23,773 (12.4)	199 (12.7)	
Shop work	9,103 (4.7)	78 (5.0)	
Nursing home	5,768 (3.0)	57 (3.6)	
Healthcare sector	21,863 (11.4)	287 (18.3)	
Home care	3,827 (2.0)	52 (3.3)	
Other	44,755 (23.3)	370 (23.5)	
Exposed to COVID-19 infected person			
Yes	32,099 (10.2)	713 (28.3)	
No	212,966 (67.4)	1,208 (48.0)	
Do not know	70,968 (22.5)	598 (23.7)	<0.001

*Alcohol use: Reporting >7 units of alcohol a week for females or >14 units of alcohol for males

** Encompasses students, stay-at-home persons, out of job, long-term sick leave, retired, and persons on parental leave.

*** Occupations are counted as the percentage of seropositive among those working. Each participant can have more than one type of occupation, why the percentage sums up to more than 100.

17. I. 184. Supplementary table 11. Are these symptoms among all participants or only those who had positive test? Please make more clear

Response to Reviewer, comment 17:

Thank you for pointing this out. These are symptoms among all participants. We have revised the Figure to include numbers in each group in the facet labels. Further, the Figure Legend has also been revised to indicate that this is among all participants:

Prior to revision:

Supplementary figure 11: Proportion of persons who experienced symptoms stratified for age groups. Numbers next to bars represent percentages.

Revision:

Supplementary figure 11: Proportion of persons who experienced symptoms stratified for age groups among all participants. Numbers next to bars represent percentages. Numbers in facet labels represent total participants within each age group.

Discussion

18. I. 202-I. 217. The discussion does not really discuss the problem with only 29% of confirmed PCR cases have positive POCT. This discrepancy has to be discussed. The antibody response of IgM and IgG is found to be highest about 2-4 weeks after symptom onset and decrease afterwards. The authors argue that for seronegative, longer time had passed from a previously positive PCR test than for seropositive. However, this statement has to be discussed in relation to relevant literature, e.g., regarding the persistence of antibodies over time. I find that it seems unlikely according to the scientific literature that 70% of those with positive PCR test do not present with antibodies. Thus, the antibody test used (POCT) is a rather big limitation in this otherwise impressive study and needs more attention in the discussion. And would you expect that IgM was more prevalent than IgG over time like found in this study? Please discuss this finding

Response to Reviewer, comment 18:

Thank you for this valuable question. From September 2020 the incidence of infected people in Denmark increased, peaking in December 2020. This could explain the higher proportion of IgM positive than IgG positive found in this study. The first infection wave in spring 2020 was minor, fewer were therefore infected back then, resulting in fewer with IgG antibodies and more with IgM antibodies during the study period. Other studies have shown waning antibody levels within several months after infection, and the diagnostic testing window is therefore of importance as the study was performed seven to eight months after the first COVID-19 case in Denmark.

We agree that the seroprevalence was lower-than-expected, with a confirmation of a positive POCT for only 29% of PCR positive. One reason could be due to the POCT performance, and other studies have also seen a lower-than-expected sensitivity for the POCT used. But the low seroprevalence could also be due to the complexity of performing and reading the POCT at home, since 2.9% were inconclusive.

We agree that the POCT used is a limitation of the study. The sensitivity of the POCT used was lower than anticipated. The major importance of the study was the identification of several factors and risk associations nationwide and across multiple subgroups. We have added this to the “Strengths and limitations” section.

Prior to revision:

POCT in general have a lower diagnostic performance compared to laboratory testing (17).

Revision (page 12, l. 207-209):

POCT in general have a lower diagnostic performance compared to laboratory testing (17) and the Livzon POCT have been found to have a lower-than-expected sensitivity (18,19).

Reference 18, 19:

Conte DD, Carvalho JMA, de Souza Luna LK, Faico-Filho KS, Perosa AH, Bellei N. Comparative analysis of three point-of-care lateral flow immunoassays for detection of anti-SARS-CoV-2 antibodies: data from 100 healthcare workers in Brazil. *Braz J Microbiol.* 2021;52(3):1161-5.

Nilsson AC, Holm DK, Justesen US, Gorm-Jensen T, Andersen NS, Ovrehus A, et al. Comparison of six commercially available SARS-CoV-2 antibody assays—Choice of assay depends on intended use. *Int J Infect Dis.* 2021;103:381-8.

Prior to revision:

The diagnostic testing window is also of importance as the study was performed seven to eight months after the first COVID-19 case in Denmark.

Revision (page 13, l. 213-216):

The diagnostic testing window is also of importance as the study was performed seven to eight months after the first COVID-19 case in Denmark, as studies have shown waning antibody levels within several months after infection (21).

Reference 21:

Arhipova-Jenkins I, Helfand M, Armstrong C, Gean E, Anderson J, Paynter RA, et al. Antibody Response After SARS-CoV-2 Infection and Implications for Immunity : A Rapid Living Review. *Ann Intern Med.* 2021;174(6):811-21.

Added to revised manuscript (p. 12, l.216-219):

From September 2020 the incidence of infected people in Denmark increased, peaking in December 2020. This could explain the higher proportion of IgM positive than IgG positive found in this study. The first infection wave in spring 2020 was minor, fewer were therefore infected back then, resulting in fewer with IgG antibodies and more with IgM antibodies during the study period (22).

Reference 22:

Hansen CH, Michlmayr D, Gubbels SM, Molbak K, Ethelberg S. Assessment of protection against reinfection with SARS-CoV-2 among 4 million PCR-tested individuals in Denmark in 2020: a population-level observational study. *Lancet.* 2021;397(10280):1204-12.

Added to revised manuscript in the “Strengths and limitations” section (p. 17, l. 323-325):

The sensitivity of the POCT used was lower than anticipated. The major importance of the study was the identification of several factors and risk associations nationwide and across multiple subgroups.

19. I. 207. You state: 'The diagnostic testing window is also of importance as the study was performed seven to eight months after the first COVID-19 case in Denmark.'. But how do you expect this to influence your results? Please elaborate

Response to Reviewer, comment 19:

Thank you for this comment. See answer in "Response to Reviewer, comment 18.

20. I. 216. I might have missed this information but where is the data regarding seropositivity among participants who did not previously have a previous positive PCR test? One could also argue that this was true among those with a positive PCR test as one would expect this number to be much higher.

Response to Reviewer, comment 20:

Unfortunately, we only have data regarding participants with a previous positive PCR test, but not regarding previous negative PCR test.

21. L. 225. Please discuss in connection with the other Danish studies why your estimate is lower both the population based and the convenient sample study.

Response to Reviewer, comment 21:

Thank you for this question.

In line 228-229 we wrote " In other Danish studies, the tests (POCT and ELISA) have been performed and read or analyzed by professional staff which increases the performance of the test." This is probably one of the reasons for the lower estimate of the seroprevalence in our study. The two mentioned Danish studies had a selected group of participants which can also be the case for the indifference in seroprevalence.

Added to revised manuscript (page 13, l. 237-240):

The discrepancy between the estimates in this study and those mentioned in earlier Danish studies may partly be due to test performance and the selection of participants. As mentioned, the earlier Danish studies were performed and analyzed by professional staff and the participants were from selected groups.

22. I. 231. Sending our test material to participants at home, may enabled inclusion of vulnerable and elderly susceptible to infection who otherwise would not have participated. But what about those, e.g., maybe elderly, who may have difficulties to perform the test at home. They may e.g. not be as liable to access the demonstration online. Could this be one reason for the lower participation among the elderly? Please include in the discussion

Response to Reviewer, comment 22:

Thank you for this interesting comment. There is a limitation in participation as the test was performed at home by participants, and this could be one of the reasons for the lower participation among the elderly in both answering the questionnaire and performing the POCT. This could probably have been optimized if project staff were able to help with the performance either by visiting people at home or in their local community, as older people may have difficulties accessing the online demonstration. We tried to overcome this by sending a thorough manual along with the POCT , with text and pictures of how to perform the POCT.

Prior to revision:

Sending our test material to participants at home, may enabled inclusion of vulnerable and elderly susceptible to infection who otherwise would not have participated.

Revision (page 14, l. 244-248):

Sending our test material to participants at home for self-test may have prevented participation of vulnerable and /or older people susceptible to infection, as the test-setup required online access to read the invitation by e-Boks as well as sending answers to the questionnaire and POCT result. The complexity of performing and reading the test could also have been a factor in low participation rate among participants in the older age group.

23. I. 240. Why do elderly test less frequently? Does e.g., distance to testing setting play a role?

Response to Reviewer, comment 23:

Thank you for this valuable comment. In October 2020 it was only possible to have a PCR test performed at a hospital or at test facilities in the larger cities. This could be a reason why older people was tested less frequently.

Added to the revised manuscript (page 14, l. 260-263):

During the study period, it was only possible to have PCR tests performed at hospitals or test facilities in the major cities. It may thus have been more difficult for older participants to be tested. Test facilities increased in Denmark during autumn/winter 2020 (22).

Reference 22:

Hansen CH, Michlmayr D, Gubbels SM, Molbak K, Ethelberg S. Assessment of protection against reinfection with SARS-CoV-2 among 4 million PCR-tested individuals in Denmark in 2020: a population-level observational study. Lancet. 2021;397(10280):1204-12.

24. I. 253. Have others reported higher antibody levels among elderly? What do you mean by distribution of antibodies? Literature regarding elderly and symptoms need to be addresses in connection with this notion that 'As such, elderly participants may more often be subject to asymptomatic infections, thereby constituting an important subgroup that may warrant further attention'.

Response to Reviewer, comment 24:

Thank you for this comment. We have made the following changes:

Prior to revision:

Nevertheless, the distribution of antibodies (comparable levels of IgG and IgM) was highest among elderly participants although they reported fewer symptoms and had fewer tests.

Revision (page 15, l.271-273):

Nevertheless, the estimate of antibodies (comparable levels of IgG and IgM) was highest among elderly participants although they reported fewer symptoms and had fewer tests.

Added to the revised manuscript (page 15, l.273-276):

This had not been seen in other Danish studies, where seroprevalence was highest among the younger age groups (5,8), except in a study of social housing areas where the seroprevalence was found increasing with age (12). As such, elderly participants may more often be subject to asymptomatic infections, thereby constituting an important subgroup that may warrant further attention.

Reference 5,8,12:

Espenhain L, Tribler S, Jørgensen CS, Holm Hansen C, Wolff Sönksen U, Ethelberg S. Prevalence of SARS-CoV-2 antibodies in Denmark 2020: results from nationwide, population-based sero-epidemiological surveys. medRxiv. 2021:2021.04.07.21254703.

Erikstrup C, Hother CE, Pedersen OBV, Molbak K, Skov RL, Holm DK, et al. Estimation of SARS-CoV-2 Infection Fatality Rate by Real-time Antibody Screening of Blood Donors. Clin Infect Dis. 2021;72(2):249-53.

Fogh K, Eriksen AR, Hasselbalch RB, Kristensen ES, Bundgaard H, Nielsen SD, et al. Seroprevalence of SARS-CoV-2 antibodies in social housing areas in Denmark. medRxiv. 2021:2021.05.07.21256725.

25. I. 256 to 258. This sentence regarding working age and attending PCR testing - is that in relation to the section I. 240-245. If yes, please move the section above. If not, please make more clear

Response to Reviewer, comment 25:

The reviewer is correct. To avoid confusing readers, we have moved the section.

Prior to revision:

Younger participants may be more exposed to infection by having more social contacts or via their employment.

Revision (page 14, I. 255-258):

Younger participants may be more exposed to infection by having more social contacts or via their employment, and it should be noted that individuals in the working age who were unable to work from home may attend PCR testing more often than people who have retired, which could contribute to our observations.

26. I. 287. Yes, serological surveys with a representative sample of the wider population are important. But is this the case in this study? Please argue for this statement

Response to Reviewer, comment 26:

Thank you for this question. The seroprevalence result was somewhat hampered by a lower-than-expected performance of the POCT and we cannot rule out selection bias, but with the use of e-Boks, an online platform, we managed to reach out to a large proportion of the Danish population. With this study we got a representative sample of the population according to age, gender, geography, occupation, socioeconomic status, ethnicity as well as geographic distribution.

27. I. 293. What about the difficulty to perform the antibody test? It should be added as a potential source of bias

Response to Reviewer, comment 27:

Thank you for this comment. This is added in the "Strength and limitation" section. See "Response to Reviewer", comment 28.

28. I. 298. I think that the antibody test used must be discussed as a limitation along with the fact that only 29% of those with positive PCR had positive POCT. Further, there is a need to address the low participation rate and representativeness of the sample.

Response to Reviewer, comment 28:

Thank you for this suggestion. We have added this to the “Strengths and limitations” section.

Prior to revision:

The low seroprevalence at 0.79% in our study is a clear limitation.

Revision (page 17, l.316-323):

The low seroprevalence at 0.79% in our study is a clear limitation and may be due to a low sensitivity of the POCT used or problems performing or reading the test results. The participant rate of 22% could be due to the requirements of online access, as invitation for participation was sent by e-Boks, an online platform, where participants sign up, answer questionnaires and report POCT results. This could probably have been optimized if not only online access was required for participation and if project staff were able to help with the performance either by visiting people at home or in their local community. The complexity of participants performing and reading the POCT could also have been a factor in the participation rate, however, with a total number of 318,522 participants.

29. I. 305. This statement is rather overall. More emphasis in which way?

Response to Reviewer, comment 29:

Thank you for this suggestion.

Regarding policy making and public health prevention, important elements in the future strategy of preventing outbreaks of infection could be easier access to test facilities and detailed information to the public on knowledge of transmission. In this study we found that although the seropositivity increased with age, participants 61 years and over reported fewer symptoms and were tested less frequently. This could be due to the difficulties of nearby tests in the period this study was conducted. A key element in identification of outbreaks and infected individuals are tests, either antigen or antibody. Focus on access to test facilities closer to ones homes or work would therefore increase the availability for tests in the broader population, and could be a way to decrease the infection transmission in local communities and households.

Prior to revision:

More emphasis should be placed on occupation, exposure in local communities as well as age of participants.

Revision (page 17, l. 330-332):

We found that more emphasis from national and local authorities towards the risk of infection should be placed on age of tested individuals, type of occupation, as well as exposure in local communities and households.

Table 1.

30. Male (%) 113,412 (422) - should be 42.2% I assume.

Response to Reviewer, comment 30:

The reviewer is correct. We have amended the typo.

31. Fig 3. Mentioning subset in the title, you mean subset based on the categories on the Y-axis?

Response to Reviewer, comment 31:

We apologize for the confusion. The word “subset” is redundant in this setting and has been removed. The categories *work/study, family/friend, >15 min. in room, body contact, household* comprises 32,812 participants exposed to COVID-19 infected persons in these settings. The risk of seropositivity was then compared to participants not exposed to COVID-19 infected persons (*Non-exposed* category).

The figure title has been revised to:

Figure 3: Risk ratio for seropositivity of 32,812 participants exposed to COVID-19 infected persons in various settings. For each setting, participants exposed to COVID-19 infected persons was compared to participants not exposed in this setting (reference group).

32. Suppl fig 7. Why is n=804?

Response to Reviewer, comment 32:

Thank you for this comment. In the submitted version it was not clear how this number is achieved. This is the number of seropositive participants of whom had both an available date of PCR and POCT test. This allowed us to investigate the proportion of seropositive participants according to time since positive POCT test. We have clarified this in the manuscript with the following revisions:

Added to revised manuscript (page 9, l. 127-128, statistical analyses):

Further, for seropositive participants with an available date of POCT and PCR test, we investigated the proportion of seropositive participants according to time since PCR test.

Prior to revision:

The proportion of seropositive participants decreased with increasing time between PCR test and POCT (Supplementary figure 7).

Revision (page 10, l. 166-169, POCT findings in participants with previous COVID-19, or a positive PCR):

For seropositive participants with an available date of positive POCT test and of the PCR test (n=804), the proportion of seropositive participants decreased with increasing time between PCR test and POCT (Supplementary Figure 7).

Prior to revision:

Supplementary figure 7: SARS-Cov-2 seropositive % among 804 individuals by POCT stratified for days since positive PCR.

Revision (page 32, Supplementary Material, Figure legends):

Supplementary Figure 7: SARS-Cov-2 seropositive % among 804 individuals with an available date of POCT and available date of PCR test. The seroprevalence is stratified for days since positive.

November 12, 2021

Dr. Kamille Fogh
Herlev and Gentofte Hospital, University of Copenhagen
Dept. of Cardiology & Dept. of Emergency Medicine
Herlev
Denmark

Re: Spectrum01330-21R1 (Testing Denmark: A Danish nationwide surveillance study of COVID-19)

Dear Dr. Kamille Fogh:

Your manuscript has been accepted, and I am forwarding it to the ASM Journals Department for publication. You will be notified when your proofs are ready to be viewed.

Sincerely,

Miguel Martinez
Editor, Microbiology Spectrum

Journals Department
Supplemental Material FOR publication: Accept